# A Statistical Approach to Describe the Ripening Evolution of Sangiovese Grapes Coming from Different Chianti Classico Sub-Areas

**DOI:** 10.3390/foods10102292

**Published:** 2021-09-28

**Authors:** Alessandro Bianchi, Isabella Taglieri, Verdiana Rimbotti Antinori, Fabrizio Palla, Monica Macaluso, Giuseppe Ferroni, Chiara Sanmartin, Francesca Venturi, Angela Zinnai

**Affiliations:** 1Department of Agriculture, Food and Environment, University of Pisa, Via del Borghetto 80, 56124 Pisa, Italy; alessandro.bianchi@phd.unipi.it (A.B.); isabella.taglieri@for.unipi.it (I.T.); giuseppe.ferroni@unipi.it (G.F.); chiara.sanmartin@unipi.it (C.S.); francesca.venturi@unipi.it (F.V.); angela.zinnai@unipi.it (A.Z.); 2Marchesi Antinori S.p.A., Via Cassia per Siena, 133, 50026 Firenze, Italy; verdiana@rimbotti.it; 3Istituto Nazionale di Fisica Nucleare (INFN), Sezione di Pisa, Largo Bruno Pontecorvo, 3, 56127 Pisa, Italy; Fabrizio.Palla@cern.ch; 4Interdepartmental Research Centre “Nutraceuticals and Food for Health”, University of Pisa, Via del Borghetto 80, 56124 Pisa, Italy

**Keywords:** Chianti Classico DOCG, Sangiovese, vineyard, grapes, statistical modelling

## Abstract

In Italy, Chianti Classico identifies a territory located in the heart of Tuscany that was once known as Chianti. From the pedological point of view, the entire DOCG (Denomination of controlled and guaranteed origin) has some common features but also shows many specific features related to certain small areas that give rise to the presence of many “terroirs”. Due to the intertwining created by the alternation of valleys and hills and the different characteristics of the territory, factors such as altitude and exposure play a very important role in the vegetative and productive expression of grapes. Some production areas were identified within the appellation where it is argued that the terroir and the grapes are quite distinct from those of other surrounding areas, albeit within the Chianti Classico appellation. On the basis of this information and considering that no data are available in the literature, the present study proposed an innovative multidisciplinary approach (analytical and statistical) that was capable of carrying out an objective evaluation of the various sub-areas investigated, using Sangiovese grapes as the variety in question. This research took into account the climatic results and the different pedological characteristics, evaluating the evolutionary phenomena that were linked to the ripening of the grapes in each phase of its formation.

## 1. Introduction

In the world when people talk about the famous wine regions [1], they always refer to a specific wine production area. Each sub-region has characteristics that distinguish it from the adjacent one, not only in terms of soil but also in terms of climate [2,3,4].

Viticulture is, therefore, one of the few activities that everywhere, in distant and more recent times, have engaged humans, who have stubbornly made even the most inhospitable and hostile areas productive, creating at the same time, especially in Italy, the most superb landscapes, so much so that the strong visual impact of these growing areas has now become synonymous with the high quality and cultural values of wine [5,6,7].

In Italy, the Chianti area is located between Florence and Siena. It is mostly known for the production of high-quality red wines, which are distinguished with a PDO label at the European level [8] and DOCG (Controlled and Guaranteed Designation of Origin) at the national level [9]. Nowadays, the production of Chianti and Chianti Classico wines was estimated to be 741,000 hL/year and 244,000 hL/year in 2019, respectively, with a real value of these productions of 91 million EUR for Chianti and 68 million EUR for Chianti Classico [10,11].

From the pedological point of view, the whole DOCG has some common traits but also shows many specificities that are linked to certain small areas that give rise to the presence of many “terroirs”, and therefore different wines. To date, the terroir concept is not easily defined and remains one of the most debated issues in the world of wine because of the large variety of interacting natural and human factors, on which there is not always agreement [12].

Indeed, it was observed that the elemental composition of grapes and wines depends on several factors, including the soil characteristics, type of grape, area of production and environmental conditions, allowing for definitions in terms of a representative “fingerprint”, which is especially important for quality wines produced in specific regions [13,14,15].

In the Chianti Classico area, the alternation of valleys and hills and the different characteristics of the territory factors, such as altitude and exposure, all play very important roles in the vegetative and productive expression of the grape [12,16,17].

Among all the Italian grape varieties, Sangiovese is one of the most important cultivars from the economic point of view, as it is widely cultivated throughout the country to produce both DOCG/DOC wines and IGT wines, while at the European level, it ranks tenth [18,19]. At least 80% of the grapes used in the Chianti Classico wine must be produced in the defined production area and the vineyard must be at least 80% Sangiovese.

A recent study conducted on the influence of the soil on the quality of Sangiovese grapes in the Chianti Classico area showed a relationship between the sugar content and nature of the grapes and soil composition, in particular, the organic and clay content [20].

Through various interviews that were carried out at the beginning of 2019 with various producers of the appellation, some production areas were identified within the appellation where it is argued that the terroir and the grapes are quite distinct from those of other surrounding areas, albeit within the Chianti Classico appellation. On the basis of this information and considering that no data are available in the literature, the present study proposed an innovative multidisciplinary approach (analytical and statistical) that was capable of carrying out an objective evaluation of the various sub-areas investigated. The research took into account the climatic results and the different pedological characteristics, evaluating the evolutionary phenomena that were linked to the ripening of the grapes in each phase of its formation.

## 2. Materials and Methods

### 2.1. The Nine Selected Areas

The study was conducted in 2019 on nine Sangiovese vineyards (*V*. *vinifera*) (Figure 1). The experimental analyses were statistically evaluated during the ripening phase in order to identify which areas were most distinguished from the others. The great impact of altitude needed to be considered since it is able to influence both the average temperatures and thermal excursion, and it is responsible for the different evolutionary kinetics of the various parameters that characterise the grapes.

The vineyards were all managed organically with only manual passage to remove any shoots grown from the old wood. To contain the development of the canopy in all the vineyards, two mechanical trimmings were carried out; moreover, light defoliation on the shaded side of the row was also carried out to allow for good aeration of the grapes. Treatments based on copper and sulphur hydroxide or tribasic copper and sulphur sulphate were necessary to manage the development of *Oidium* and *Peronospora* diseases during the season. As for the control of moths, the method of sexual confusion was used in all vineyards. The management of the vineyards made it possible to obtain healthy plants overall, and plants were rejected in the case of slight attacks or damaged clusters, paying close attention to the selection of the samples to be used.

Table 1 shows the agronomic criteria, while Table 2 shows the characteristics of the selected vineyards.

### 2.2. Description of the Weather Station

MeteoSense 2.0 (NetSens) meteorological control units located near the vineyards that were selected for the study were used to monitor the weather conditions of each vineyard. These weather stations were equipped with several sensors that allowed for the instantaneous detection of air temperature, humidity, leaf wetting and the intensity and amount of rain. The data recorded in the field by the stations were automatically sent to a LiveData cloud portal, which allows for access from any device, e.g., computer or cell phone. The data that are recorded in the cloud allow for the formation of a historical database of great importance that can be used for the evaluation of the meteorological trend of the different years and areas.

For this study, the weather stations were initially used for the evaluation of weather conditions on the day of grape sampling, which was necessary for the evaluation of the maturity of the vineyard. This was done in order to ensure fairly similar sampling conditions and thus avoid a situation where the analyses were not comparable. Subsequently, the meteorological archive of the system was used to evaluate the different climatic trends of each area with particular reference to August to October, which are those months of greatest interest as they coincide with the period of major importance of the plant. In order to compare all nine vineyards, the following parameters were selected: average temperature, average excursion, maximum of the maximum temperature and minimum temperature, rainfall and relative humidity.

### 2.3. Soil Characteristics

The Chianti Classico region covers almost 72,000 hectares between Florence and Siena. The territory can be assimilated to a rectangular region, bounded by the Chianti Mountains that constitute its eastern border [22].

The soil skeleton consists mainly of two stones type: Galestro and Alberese. The first is a rock that is very typical of the Apennines and consists of hard clayey schist but is very easy to separate. With the addition of water, this stone tends to dissolve and become mud that regains its hardness once it is returned to dryness [23]. Alberese, on the other hand, is a greyish-white material of a marly limestone nature (over 80% is made up of calcium carbonate) that is relatively localised in some areas of Chianti Classico and is a material that is considered to be very resistant to atmospheric agents [23]. In contrast, the soil at the base of the hill is generally rich in clay. Clay has an important function in the constitution of the water reserves of the soil; often, if it is too high, it leads to the formation of underground stagnations that are responsible for the development of an anoxic environment in which the roots of plants have difficulty surviving. Despite this, in the lower part of the hills, the soils tend to be cooler and richer in nutrients; moreover, with the thermal sums being higher than in the higher areas, the ripening of the grapes is more anticipated [24].

The Chianti Classico territory can be divided into five areas (Figure 1) [21,24,25].

Area A, located on the east side of the DOCG, has an average altitude of 450 m a.s.l. In these areas on sandstone rock (Macigno del Chianti), the abundant soil skeleton consists mainly of Galestro, with the exception of the Radda area, where it is possible to identify some points containing Alberese (mostly west of Radda). In addition to the abundant content of the soil skeleton, 20–25% clay is present: this mixture guarantees a good drainage capacity and, at the same time, also allows for good water reserves. The higher areas of this region have soils that tend to be poorer and less deep, making them capable of delaying the vegetative development and ripening of the grapes. Near the town of Gaiole, especially on the eastern side, the pH of the soils is more prone to neutrality.

Area B is located in the heart of Chianti Classico, where the hillsides drop (about 350 m a.s.l.) while moving towards the west. The soils in this area are mainly calcareous-clayey, with a moderately alkaline soil pH. Furthermore, in the area of Panzano, it is possible to find Pietraforte as the main soil skeleton component (brown sandstone made of carbonate cement inside and very appreciated for its resistance).

Area C corresponds to the municipality of San Casciano and is therefore located in the northern area of the Chianti Classico. The altitudes continue to decrease (on average 290 m a.s.l.); therefore, the temperatures are slightly higher and the soil tends to be cooler. In these localities, the soils are based on marine deposits with a prevalence of sand. Consequently, on the surface of the soils, it is possible to observe a skeleton that is based on siliceous pebbles (round white stones), which contribute to the fairly early ripening of the grapes.

Area D, located to the south, corresponds to the territories of Castelnuovo Berardenga. Here the soils vary from sandstone rocks (especially on the right side), predominantly sandy marine deposits (mostly in the central part), predominantly clayey marine deposits (particularly on the left side), and mainly calcareous-clayey rocks (a little on all sides). Being in the southernmost part of the Chianti Classico, Sangiovese tends to be produced more abundantly and matures earlier here than in the other areas.

Finally, area E, located on the south/west border, consists of deep soils on ancient lake deposits. The lower altitudes (about 230 m a.s.l.) imply a greater presence of clay in the soil and a smaller presence of a mainly tuffaceous skeleton.

Table 3 summarises the topographical characteristics and soils of the selected vineyards.

### 2.4. Description of Sampling Modes

The sampling of the grapes began in August 2019. In order to identify the degree of ripeness of the vineyard, 60 plants per vineyard were selected and marked out of 8 rows used as “indicator plants”, which were sampled about every 6–8 days until the harvest day. Since these plants had to be representative of the chosen vineyard, it was considered important not to select individuals placed at the edge of the vineyard, they had to be at least 5 m from each other, have a vigour that was in line with the surrounding plants and be pathologically healthy.

The grapes were sampled by taking 100 random berries from both the individual vines and from various sides and parts of the bunch. Once packed and labelled, they were brought to the laboratory for routine analysis to assess the ripening trend, which was essential to identify the ideal technological ripening point and therefore also the harvest date for each of the nine areas.

### 2.5. Chemical Composition Analyses

All chemical determinations that were necessary for the characterisation of grapes (sugar content (hexoses g/L), titratable acidity (tartaric acid g/L), pH, L-malic acid (g/L), tartaric acid (g/L) and potassium (mg/L)) were performed following the OIV methods [26]. In particular: a refractometer was used for the sugar determination according to the OIV-MA-AS2-02 method; the determination of titratable acidity was carried out according to the OIV-MA-AS313-01 method with a potentiometric titration; for the determination of the pH, a pH meter was used according to the OIV-MA-AS313-15 method; to determine the content of L-malic acid, the OIV-MA-AS313-11 enzymatic method was used; tartaric acid was precipitated in the form of calcium (±) tartrate and determined gravimetrically according to the OIV-MA-AS313-05A method; potassium content was determined directly in diluted wine using atomic absorption spectrophotometry after the addition of cesium chloride to suppress the ionisation of potassium according to the OIV-MA-AS322-02A method.

### 2.6. Statistical Analyses

In order to compare the nine vineyards, the program Root Data Analysis Framework based on C++ language was used for the data processing [27,28]. The program uses a minimum model 2 that allows for the following function to be reduced to a minimum:(1)$$\chi^{2}=\sum_{i=1}^{N}\frac{{\left(O_{i}-E_{i}\right)}^{2}}{\sigma_{i}^{2}}$$
where $O_{i}$ is the ith measurement, $E_{i}$ is the predicted value of the observable at the same date of the measurement and $\sigma_{i}^{2}$ is the uncertainty of the measurement. The sum traces all measurements for the observable data. The measurements were plotted as a function of time so that the different areas could be compared.

## 3. Results and Discussion

### 3.1. Climatic and Vineyard Site Considerations

The climatic parameters are represented in Table 4. The data used were obtained by processing the values that were recorded by the meteorological stations located near the selected vineyards, as previously described (Section 2.2). From the analysis of the parameters shown in Table 4, a remarkable difference emerged between the low rainfall (0–50 mm) relative to some vineyards (Panzano, San Casciano and Castelnuovo Berardenga) as opposed to the significantly more abundant rainfall (larger than 90 mm) recorded in other vineyards (Castellina, Radda and Tavarnelle). As far as the relative humidity values were concerned, the various areas showed more contained differences, so much so that more than half of the vineyards (Castellina, Gaiole, Greve, Radda and Tavarnelle) had values above 70%, and the others were between 50 and 65%. A substantial homogeneity of the average temperature values was observed between the different areas. In fact, most of the vineyards had an average temperature between 20 and 22 °C, with the exception of San Casciano and Greve, which were warmer with temperatures of 23.7 and 22.5 °C, respectively, and Radda and Lamole, which were much cooler with temperatures of 19.2 and 19.5 °C, respectively. A remarkable difference between the maximum (T.max) and the minimum (T.min) daily temperatures emerged between the vineyards. In fact, San Casciano had the highest values in both parameters (38.4 and 22.6 °C, respectively) and, together with the lowest excursion and reduced rainfall, was the hottest vineyard and therefore had the greater stress. In contrast, Radda and Lamole showed the lowest values of these parameters and the highest excursions, which were possibly explained by the fact that their altitudes (more than 440 m a.s.l.) favored a cooler climate. The remaining vineyards were placed in an intermediate position, with values of T.max and T.min ranging between 34 and 37 °C and between 19 and 21 °C, respectively. Finally, looking at the differences between the average excursion data, we noticed a more pronounced variability: from about 10–12°C for San Casciano and Greve up to excursion values of even 15–16 °C (Radda 16.6 °C, Lamole 16.2°C and Castelnuovo Berardenga 15.4 °C).

Another very relevant variable for the ripening of the grapes is the altitude of the vineyard, with the main advantage being that vineyards on slopes at a higher altitude receive more solar radiation and benefit from cool temperatures, particularly at night, thus slowing down the grape-ripening process, which in turn increases the production of flavour compounds in the skin. At the same time, the sugars ripen and the acidity decreases: this behaviour tends to happen slowly, thus the grapes are often harvested with fully ripe flavours before the sugar levels have risen to values that would make the wine very alcoholic (more than 14.5% *v*/*v*). Besides these conditions, high-altitude grapes tend to have a more pronounced and attractive acidity compared to those in the plain [29,30]. Each area, characterised by considerable differences from the pedoclimatic point of view, will have potential repercussions on the compositional characteristics of the grapes.

### 3.2. Evolution of the Composition of the Grapes during the Pre-Harvest Period

The trends of the main parameters that contributed to identifying the technological maturity of the grapes during the 2019 vintage were evaluated. The grapes were harvested when the parameters that characterised their technological maturity (in particular, sugar content, pH, titratable acidity, malic acid content) reached values that are appropriate for the type of wine to be produced. The analytical data were evaluated together with the trends of the lines that best correlate with the evolutionary curve of the parameter, in this case, the equation used was:(2)y=mt+q
where *y* is the parameter of interest and *t* is the time to harvest, which varied from one area to another; thus, the analytical time considered took into account the day of the veraison (*t* = −40 days) in order to allow for the comparison of the different sub-areas.

#### 3.2.1. Grapes’ Primary Metabolism: Accumulation of Sugars

Table 5 shows the parameters of the equation, while Figure 2 shows the graphical trend of the sugar accumulation in order to compare the trends for the different areas examined. Indeed, even if the sugar content increased with time (all the slopes were positive), the evolutionary kinetics was different and we could divide the areas into three groups depending on the rate of accumulation of sugars: slow (Gaiole and Radda), intermediate (Castellina, Lamole, San Casciano, Tavarnelle, Castelnuovo Berardenga, Panzano) and fast (Greve). The data reported show a difference in the initial sugar content (*t* = −40 days) of the grapes from the different vineyards: the grapes with the lowest sugar content at the beginning of the sampling (San Casciano and Greve) recorded the highest sugar content at harvest time. Instead, the grapes with the highest sugar content at the start of the sampling (Gaiole and Radda) had the lowest sugar content at the harvest date, excluding Lamole. An exception was the case of Lamole, which, despite having started with high sugar content, did not follow the evolutionary effect of the other areas since it started and remained in the high category. The intermediate zones, on the other hand, had accumulated a small sugar content, thus exhibiting a low final content (Panzano and Tavarnelle) or managed to load a little more while remaining in the intermediate category (Castellina and Castelnuovo Berardenga).

#### 3.2.2. Grapes’ Primary Metabolism: Decrease in Acids

A similar approach, taking into account the fact that technological maturity provides for a decrease in titratable acidity (unlike what happens for sugars), was followed for the evaluation of the titratable acidity, malic acid and pH of the grapes. In order to confirm the existence of an inverse correlation between the evolution of the sugar content compared to the acid evolution, from the evaluation of the parameters of the straight lines that correlated the analytical data obtained in the period preceding the harvest, similarities and differences emerged regarding the metabolic behaviour of the grapes from the different vineyards.

Indeed, as seen in Table 6 and Figure 3, the titratable acidity of the San Casciano sample showed a considerable reduction, as represented by the slope of the straight line, so much so that the harvest value was the lowest among those obtained, indicating the analogous rapid metabolic evolution; the same was observed for the carbohydrate content. This trend, as evidenced by the climate analysis, could be linked to the warmer and drier conditions for this area. A similar evolution was found for Lamole; in fact, even if at harvest, it belonged to the high category, at the end of the ripening of the grapes, it ended up having an intermediate titratable acidity. Greve and Castellina started from an intermediate acid content, but at the end of the ripening of the grapes, they remained in the original category, showing an intermediate slope. Although Castelnuovo Berardenga and Radda began the ripening with low titratable acidity, they finished with an intermediate acidity; hence, the slope of the grafting is negative. Very interesting is the case of Gaiole, which although starting with high sugar content, did not follow a similar evolution to that of the other areas since it started and remained in the high category. Finally, Tavarnelle, which started with an intermediate acid content, had a high content on the day of the harvest, showing a not very marked slope.

Since the analysis of the titratable acidity evaluated a set of components, it was interesting to investigate the evolution of malic acid in grapes (Figure 3). Malic acid is known to be strongly influenced by the temperature of the territory. Warmer climates promote higher malic acid values, while colder climates exhibit lower amounts. High temperatures also cause a greater acid loss, as shown by the examined samples (Table 7, Figure 4).

As is clearly readable, San Casciano was the area with the highest quantity of malic acid, while Radda and Castelnuovo Berardenga were the areas with the lowest value. This was in line with their titratable acidity; indeed, malic acid was less stable than tartaric acid, making it easier to consume it for breathing or other metabolic pathways. In the presence of a large quantity of malic acid (e.g., San Casciano), a large titratable acidity value is normally expected. Similarly, the smaller the amount of malic acid (e.g., Radda), the smaller the titratable acidity value.

In accordance with what was observed for titratable acidity and malic acid, the pH of the grapes (Table 8, Figure 5) remained relatively constant, as represented by the value of the slope of the regression being practically zero. An interesting phenomenon occurs in Gaiole, where the pH remained constant during the period under analysis, but unlike the other areas, it showed a slightly higher decrease in titratable acidity, which was correlated with a greater decrease in malic acid.

With regard to tartaric acid, for all the tests examined, the values remained almost constant over time, hence no data are reported.

Potassium is another parameter that is taken into consideration. Figure 6 shows how much difference there was between the trends in the various areas. The climate and the territory have a great influence on this parameter. Indeed, the potassium available is that coming from clayey soils rich in organic matter where the climate is not subject to excessive rainfall, which would cause runoff. This was the case in Greve, San Casciano, Panzano and Castelnuovo Berardenga, which showed higher potassium values at the harvest (Table 9). On the other hand, sandy soils, which are poor in organic matter and characterised by regular rains, have lower values of this product. This was the case in Gaiole and Radda, which recorded the lowest values. All the other areas (Castellina, Tavarnelle and Lamole) had intermediate potassium values since they were the result of compensatory effects between climatic and territorial parameters (Table 9).

## 4. Conclusions

Chianti Classico is a territory famous for its strong vocation for the production of wine. The nine areas considered in this work are naturally considered to be different from each other and were selected to study whether the differences between the regions that are considered evident can be explicitly found in the quality of the grapes.

From the climatic analysis carried out, it emerged that there were rather important thermal differences between the nine selected areas, both in terms of the average temperature and the excursions and precipitation. The areas of Radda and Lamole had lower temperatures and excursions due to the different altitudes of the vineyards (Lamole 546 m a.s.l. and Radda 449 m a.s.l.). San Casciano and Castelnuovo Berardenga instead stood out for their high average temperatures, albeit with different temperature ranges (low for San Casciano and high for Castelnuovo Berardenga). Furthermore, the rainfall in the different areas of the Chianti Classico varied according to their position, resulting in a different water state of the plants during the ripening period of the grapes.

All grapes showed similar ripening trends of the different analytical parameters, although with different dynamics. For example, sugars during the sampling period increased for all areas, as expected. A novelty that this work showed for this parameter was in terms of the dynamics of the areas located at the extremes of sugar content: in fact, the areas that at the beginning of the sampling had a high sugar content (Gaiole and Radda) showed a low growth rate and finished the ripening with a lower sugar content than the other areas; in contrast, those areas that started at the beginning of the sampling period with very low sugar content (San Casciano and Greve) showed a fast increase in the growth rate, resulting in grapes with the highest sugar content at harvest. The only anomalous case, however noteworthy, was the one of Lamole, which even though it started the sampling with high sugar content, at harvest, it was found to be the grape with the highest sugar content of all areas, therefore establishing a good accumulation rate.

Furthermore, for the titratable acidity, a phenomenon similar to that previously explained occurred, even if the general trend was of decreasing acidity during the ripening period, as expected. Here the vineyards that showed very strong dynamics (San Casciano and Panzano) had a strong drop at harvest, while the grapes that started with low total acidity (Castelnuovo Berardenga and Radda) showed good acidity at harvest when all nine areas were compared.

Regarding the malic acid, it is known that its content and its course during the grape-ripening period depend very much on the climatic characteristics of the area where the plant grows; thus, looking at the climatic data, it is easy to understand the reason why there was a greater reduction in San Casciano than in Radda and Gaiole, which had smaller decreases.

The potassium parameter was the one that was most affected by the influence of the climate–soil interaction on grape ripeness. While more clayey soils slowed down its degradation, in contrast, cooler climates and higher rainfall intensified the loss.

In conclusion, this study represents the first step in the conceptualisation of the viticultural aspects of the area considered. The idea was to clarify which factors characterised particular production areas and, consequently, which practical aspects were linked to them in order to create a predictive model that can be used by winemakers for their productions. Furthermore, the division into sub-areas could be an easy solution to decrease the competitiveness of small producers and strengthen the local economy.

## Figures and Tables

**Figure 1 foods-10-02292-f001:**
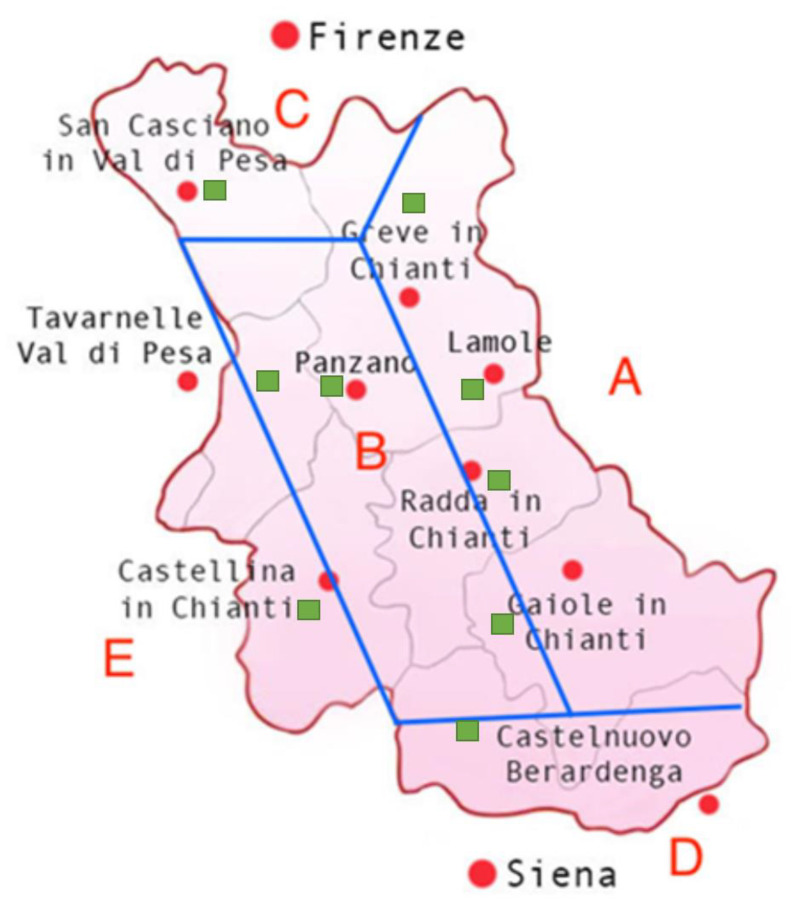
Topographical breakdown of Chianti Classico. The letters identify the soil composition of each area according to [21]. The approximate positions of the municipalities located within the area under examination are indicated in red. The approximate positions of the vineyards selected for the project are highlighted in green.

**Figure 2 foods-10-02292-f002:**
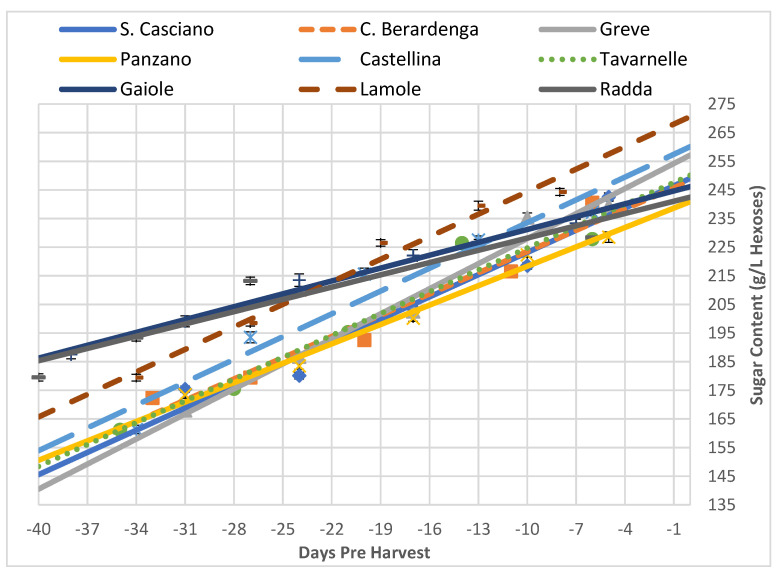
Trends of the sugar concentration in all the analysed vineyards as a function of time (expressed as negative days before the harvest). Each analytical point is reported with its 68% C.L. interval.

**Figure 3 foods-10-02292-f003:**
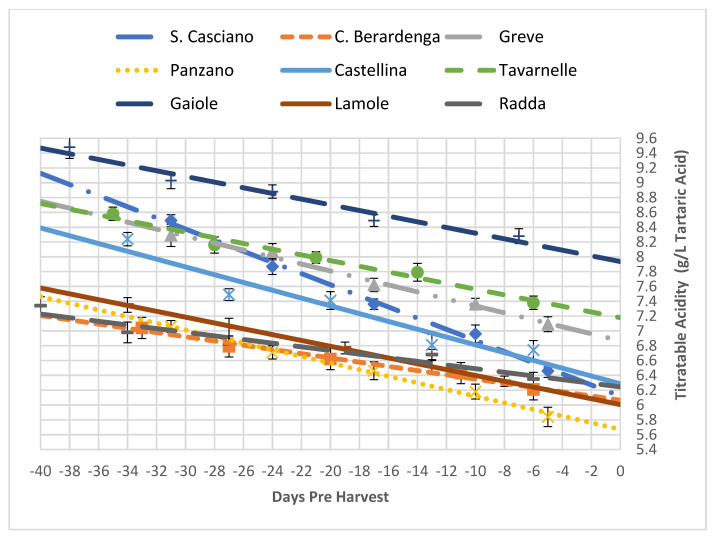
Trends of the titratable acidity in all the analysed vineyards as a function of time (expressed as negative days before the harvest). Each analytical point is reported with its 68% C.L. interval.

**Figure 4 foods-10-02292-f004:**
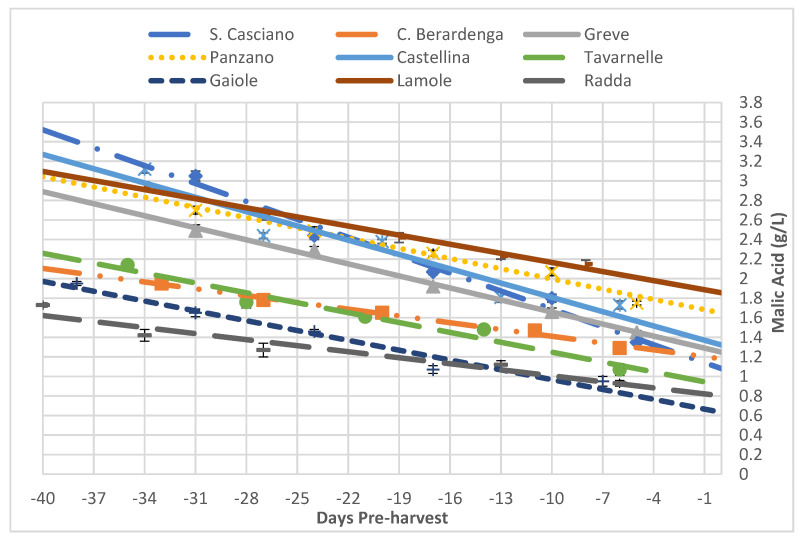
Trends of the malic acid in all the analysed vineyards as a function of time (expressed as negative days before the harvest). Each analytical point is reported with its 68% C.L. interval.

**Figure 5 foods-10-02292-f005:**
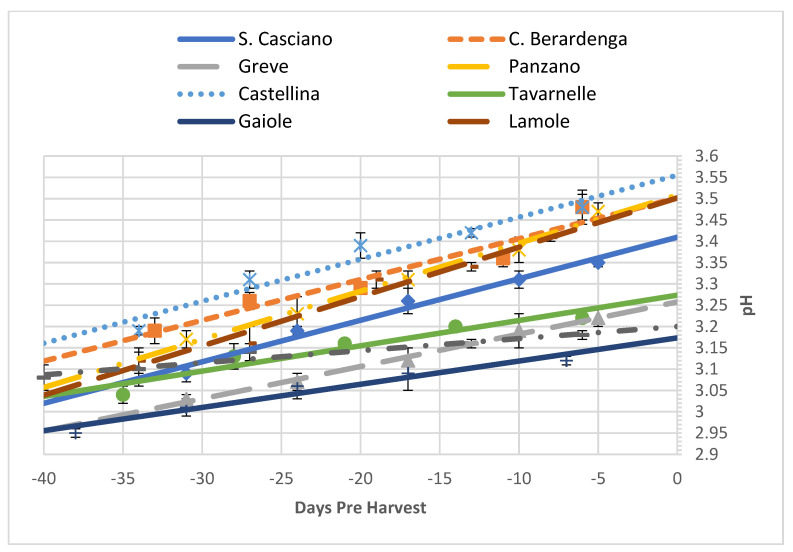
Trends of the pH in all the analysed vineyards as a function of time (expressed as negative days before the harvest). Each analytical point is reported with its 68% C.L. interval.

**Figure 6 foods-10-02292-f006:**
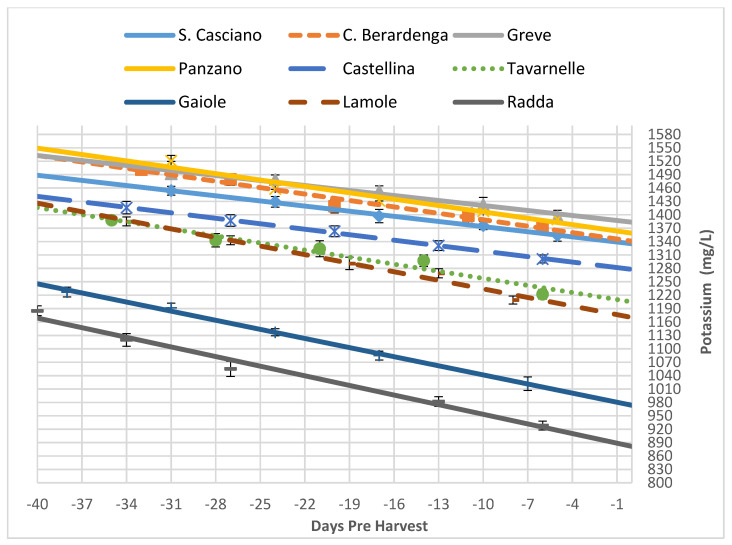
Trends of the potassium in all the analysed vineyards as a function of time (expressed as negative days before the harvest). Each analytical point is reported with its 68% C.L. interval.

**Table 1 foods-10-02292-t001:** Agronomic parameters of the chosen vineyards.

Agronomic Parameters	
**Variety**	100% Sangiovese
**Clones**	As genetically similar as possible
**Rootstock**	As similar as possible
**Planting distances**	80–90 cm × 250–280 cm
**Plant density**	Around 4500 to 5000 plants per hectare
**Training system**	Spurred cordon 80–90 cm
**Age of the vineyard**	Between 12 and 20 years
**Number of vital spurs**	More than 3 have approximately 6 to 8 productive shoots
**Number of grapes**	Around 6–8 bunches
**Production**	Between 60–75 quintals per hectare
**Vigour**	Normal
**Phytosanitary management**	According to the good agricultural practices (GAPs) to produce good-quality grapes

**Table 2 foods-10-02292-t002:** Technical and agronomic data related to the vineyards that were the subject of the experimentation.

Vineyard	Location	Total Area (ha)	Planting Density (cm × cm)	Pruning System	Rootstock	Clone	Planting Year
**Castellina**	*43°* *26’* *32.8″ N 11°14′58.7″ E*	3	250 × 80	Cordon	110R	VCR 23	2005
**Castelnuovo Berardenga**	*43°23′46.1″ N 11°20′35.6″ E*	2.2	275 × 80	Cordon 30%	775P, VGV16	SG 12T	2007
**Gaiole**	*43°25′44.8″ N 11°23′16.0″ E*	2.1	250 × 80	Cordon	110R	VCR 30	2006
**Greve**	*43°40′49.6″ N 11°20′16.7″ E*	1.7	230 × 80	Cordon	779P, 1103P	SSF9A448, R24	2000
**Radda**	*43°28′46.7″ N 11°22′28.2″ E*	0.9	250 × 80	Cordon 20%	110R	VCR5	2000
**Lamole**	*43°32′49.6″ N 11°21′11.6″ E*	1.1	230 × 80	Cordon 40%	420A	GM32	2009
**Panzano**	*43°33′33.0″ N 11°16′01.8″ E*	0.4	230 × 80	Cordon 50%	110R	G 76	2003
**San Casciano**	*43°40′24.6″ N 11°10′33.4″ E*	1.3	280 × 80	Cordon	420A, SO4	Ch. 2005, Ch. 2002, R23	2009
**Tavarnelle**	*43°34′53.3″ N 11°14′46.6″ E*	2.1	230 × 80	Cordon	420A	BF30	2004

**Table 3 foods-10-02292-t003:** Topographical and soil characteristics of the nine vineyards selected.

Vineyard	Soil Classification	Altitude	Slope	Soil Exposure	Row Orientation
**Gaiole**	A (sandstone + Galestro)	444 m a.s.l.	13.79%	SW	NE–SW
**Greve**	A (sandstone + Galestro)	289 m a.s.l.	21.30%	SE	NW–SE
**Radda**	A (sandstone + Galestro)	449 m a.s.l.	18%	SE	NW–SE
**Panzano**	B (calcareous-clayey + Alberese/Pietraforte)	335 m a.s.l.	28.20%	SE	NW–SE
**Lamole**	B (calcareous-clayey + Alberese)	546 m a.s.l.	15.30%	SW	NE–SW
**San Casciano**	C (sandy marine deposits + siliceous pebbles)	281 m a.s.l.	0.60%	E	NE–SW
**Castelnuovo Berardenga**	D (marine deposits + prevalence of clay)	386 m a.s.l.	23.70%	NW	NW–SE
**Castellina**	E (ancient lake deposit + low skeleton made of tuff)	289 m a.s.l.	12.90%	N	N–S
**Tavarnelle**	E (ancient lake deposit + low skeleton made of tuff)	333 m a.s.l.	7.14%	SE	NW–SE

**Table 4 foods-10-02292-t004:** The climate parameters (average temperature (°C), average excursion (°C), max of the maximum temperature and minimum temperature (°C), rainfall (mm) and relative humidity (%)) that were recorded in the areas under investigation and the altitude of the vineyard.

Vineyard	Altitude (m a.s.l.)	Average Temperature (°C)	Max of T.max (°C)	Max of T.min (°C)	Average Excursion (°C)	Relative Humidity (%)	Rainfall (mm)
**Castellina**	289	21.8	35.8	19.9	14.3	79.9	169.5
**Castelnuovo Berardenga**	386	21.5	37.0	19.5	15.4	51.0	47.0
**Gaiole**	444	20.8	34.8	19.7	13.1	78.7	80.8
**Greve**	289	22.5	36.6	21.2	11.3	73.0	83.4
**Lamole**	546	19.5	33.3	18.5	16.8	55.2	60.2
**Panzano**	335	21.5	36.2	20.1	13.7	57.0	16.6
**Radda**	449	19.2	33.7	18.1	16.6	74.0	95.2
**San Casciano**	281	23.7	38.4	22.6	10.0	64.0	38.8
**Tavarnelle**	333	21.0	35.1	19.8	12.6	73.3	105.9

**Table 5 foods-10-02292-t005:** Calculated values of the functional parameters, intercept, slope and R^2^ of the sugar values inside the grapes coming from the different vineyards.

Vineyard	Sugar Value at Veraison (*t* = −40 days) (g/L)	q (Intercept)	m (Slope)	R^2^	Sugar Value at Harvest (g/L)
**Greve**	140.3 ± 2.8	257.1 ± 3.2	2.92 ± 0.12	0.97	247.2 ± 1.5
**Castellina**	154.1 ± 3.9	260.1 ± 4.5	2.65 ± 0.13	0.94	244.7 ± 0.8
**Lamole**	165.8 ± 2.2	270.6 ± 2.8	2.62 ± 0.11	0.97	266.8 ± 0.9
**San Casciano**	145.4 ± 1.2	249.0 ± 2.1	2.59 ± 0.14	0.95	253.7 ± 1.7
**Tavarnelle**	148.5 ± 1.8	250.1 ± 1.9	2.54 ± 0.12	0.94	239.3 ± 1.1
**Castelnuovo Berardenga**	149.4 ± 2.4	248.2 ± 2.5	2.47 ± 0.13	0.95	249.8 ± 1.3
**Panzano**	150.5 ± 3.2	240.9 ± 3.9	2.26 ± 0.16	0.98	231.3 ± 1.6
**Gaiole**	186.2 ± 1.6	246.2 ± 1.5	1.50 ± 0.11	0.98	247.2 ± 1.3
**Radda**	185.3 ± 2.5	242.5 ± 2.3	1.43 ± 0.12	0.91	239.8 ± 1.2

**Table 6 foods-10-02292-t006:** Calculated values of the functional parameters, intercept, slope and R^2^ of the titratable acidity inside the grapes coming from the different vineyards.

Vineyard	Titratable Acidity at Veraison (*t* = −40 days) (g/L)	q (Intercept)	m (Slope)	R^2^	Titratable Acidity at Harvest (g/L)
**San Casciano**	9.12 ± 0.12	6.12 ± 0.08	−0.075 ± 0.005	0.99	6.16 ± 0.09
**Castellina**	8.41 ± 0.09	6.29 ± 0.11	−0.053 ± 0.002	0.92	6.22 ± 0.10
**Greve**	8.75 ± 0.13	6.87 ± 0.09	−0.047 ± 0.003	0.99	6.90 ± 0.12
**Panzano**	7.47 ± 0.08	5.67 ± 0.06	−0.045 ± 0.004	0.99	5.70 ± 0.13
**Tavarnelle**	8.74 ± 0.11	7.18 ± 0.07	−0.039 ± 0.002	0.98	7.30 ± 0.12
**Gaiole**	9.46 ± 0.08	7.94 ± 0.08	−0.038 ± 0.004	0.96	8.09 ± 0.13
**Lamole**	7.56 ± 0.06	6.00 ± 0.11	−0.039 ± 0.002	0.99	6.17 ± 0.11
**Castelnuovo Berardenga**	7.22 ± 0.07	6.06 ± 0.11	−0.029 ± 0.003	0.98	6.17 ± 0.12
**Radda**	7.24 ± 0.11	6.24 ± 0.07	−0.025 ± 0.003	0.91	6.22 ± 0.09

**Table 7 foods-10-02292-t007:** Calculated values of the functional parameters, intercept, slope and R^2^ of the malic acid concentration inside the grapes coming from the different vineyards.

Vineyard	Malic Acid at Veraison (*t* = −40 days) (g/L)	q (Intercept)	m (Slope)	R^2^	Malic Acid at Harvest (g/L)
**San Casciano**	3.52 ± 0.03	1.08 ± 0.04	−0.061 ± 0.004	0.96	1.02 ± 0.03
**Castellina**	3.28 ± 0.04	1.32 ± 0.05	−0.049 ± 0.003	0.92	1.29 ± 0.02
**Greve**	2.89 ± 0.04	1.25 ± 0.03	−0.041 ± 0.003	0.99	1.29 ± 0.03
**Panzano**	3.05 ± 0.05	1.65 ± 0.02	−0.035 ± 0.002	0.98	1.68 ± 0.03
**Tavarnelle**	2.27 ± 0.04	0.91 ± 0.03	−0.034 ± 0.002	0.96	0.99 ± 0.04
**Gaiole**	1.95 ± 0.03	0.63 ± 0.04	−0.033 ± 0.002	0.96	0.70 ± 0.03
**Lamole**	3.10 ± 0.04	1.86 ± 0.03	−0.031 ± 0.001	0.98	2.02 ± 0.03
**Castelnuovo Berardenga**	2.10 ± 0.02	1.18 ± 0.03	−0.023 ± 0.002	0.99	1.27 ± 0.02
**Radda**	1.60 ± 0.03	0.80 ± 0.02	−0.020 ± 0.003	0.92	0.81 ± 0.03

**Table 8 foods-10-02292-t008:** Calculated values of the functional parameters, intercept, slope and R^2^ of the pH inside the grapes coming from the different vineyards.

Vineyard	pH at Veraison (*t* = −40 days)	q (Intercept)	m (Slope)	R^2^	pH at Harvest
**Lamole**	3.02 ± 0.02	3.50 ± 0.02	0.012 ± 0.001	0.97	3.47 ± 0.02
**Panzano**	3.07 ± 0.02	3.51 ± 0.03	0.011 ± 0.002	0.98	3.54 ± 0.03
**Castellina**	3.15 ± 0.01	3.55 ± 0.02	0.010 ± 0.001	0.95	3.52 ± 0.02
**Castelnuovo Berardenga**	3.10 ± 0.02	3.50 ± 0.03	0.010 ± 0.002	0.93	3.58 ± 0.02
**San Casciano**	3.01 ± 0.03	3.41 ± 0.04	0.010 ± 0.002	0.98	3.38 ± 0.03
**Greve**	2.94 ± 0.03	3.26 ± 0.02	0.008 ± 0.001	0.99	3.24 ± 0.02
**Tavarnelle**	3.03 ± 0.01	3.27 ± 0.02	0.006 ± 0.002	0.91	3.23 ± 0.02
**Gaiole**	2.93 ± 0.02	3.17 ± 0.01	0.006 ± 0.001	0.95	3.12 ± 0.01
**Radda**	3.08 ± 0.02	3.20 ± 0.02	0.003 ± 0.001	0.95	3.17 ± 0.01

**Table 9 foods-10-02292-t009:** Calculated values of the functional parameters, intercept, slope and R^2^ of the potassium inside the grapes coming from the different vineyards.

Vineyard	Potassium at Veraison (*t* = −40 days) (mg/L)	q (Intercept)	m (Slope)	R^2^	Potassium at Harvest (mg/L)
**Greve**	1532 ± 12	1383 ± 13	−3.73 ± 0.13	0.97	1374 ± 11
**San Casciano**	1488 ± 18	1335 ± 15	−3.83 ± 0.14	0.99	1345 ± 10
**Castellina**	1440 ± 13	1277 ± 17	−4.08 ± 0.18	0.99	1284 ± 8
**Panzano**	1549 ± 14	1359 ± 19	−4.75 ± 0.15	0.95	1354 ± 14
**Castelnuovo Berardenga**	1533 ± 13	1341 ± 11	−4.79 ± 0.09	0.96	1365 ± 13
**Tavarnelle**	1416 ± 16	1205 ± 15	−5.28 ± 0.15	0.95	1188 ± 13
**Lamole**	1426 ± 14	1170 ± 10	−6.40 ± 0.10	0.98	1164 ± 12
**Gaiole**	1245 ± 15	973 ± 13	−6.80 ± 0.16	0.99	1005 ± 14
**Radda**	1168 ± 12	881 ± 11	−7.18 ± 0.17	0.98	935 ± 10

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
