# Peer review of "A Statistical Approach to Describe the Ripening Evolution of Sangiovese Grapes Coming from Different Chianti Classico Sub-Areas"

_foods, 2021, doi:10.3390/foods10102292_

Round 1

Reviewer 1 Report

Manuscript presents a study where the grape quality was evaluated from the perspective of different environmental parameters such as soil, climate, rain during ripening in nine areas of Chianti Classico territory. Analytically methods concerning grape quality are routine analysis. Authors may be need to add some brief desription of analysis since not all readers have access to OIV.

Manuscript is well written and presented, with minor spelling and grammar mistakes. Conclusions are supported by results. Authors should highlight more the novelty of their research and how grape and wine producers can benefit from this work since this is the target group of this work. I would also expect authors to refer to vine diseases, how they are affect ripening and how they are treated. Please comment on this.

Author Response

Reviewer 1

Thank you so much for the time you dedicated to the revision of our manuscript. The changes you suggested have been highlighted in yellow in the text and are listed below.

Manuscript presents a study where the grape quality was evaluated from the perspective of different environmental parameters such as soil, climate, rain during ripening in nine areas of Chianti Classico territory. Analytically methods concerning grape quality are routine analysis. Authors may be need to add some brief desription of analysis since not all readers have access to OIV.

We thank the Referee for the suggestions and consequently we have added some short descriptions for each analysis (see lines 188-197).

Manuscript is well written and presented, with minor spelling and grammar mistakes. Conclusions are supported by results. Authors should highlight more the novelty of their research and how grape and wine producers can benefit from this work since this is the target group of this work.

We thank the Referee for the suggestions. We have improved the conclusions to highlight the novelty and benefits for the producer (lines 408-413).

I would also expect authors to refer to vine diseases, how they are affect ripening and how they are treated. Please comment on this.

According to the Referee’s suggestion, we have added some details about grape diseases and the vineyard management (see lines 83-92).

Reviewer 2 Report

Purpose of the paper is to use analytical and statistical approaches to evaluate Sangiovese grapes. Utilized a multidisciplinary approach looking at characteristics associated with ripening. 

Line 83 - 84: The sentence structure needs some work.

Figure 1: Is it possible to provide/use a clearer picture. It's difficult to read the various cities on the map. I know you describe the differences between the different regions above and below the figure. Is it possible to give key differences about the different regions in the subtext of the figure.

What type of statistics did you run. You briefly talk about the program language. 

Eq. 1: Is  the equation supposed to be y = mx+b. As written I am confused by what is happening between m and t. 

Line 248 - 251: The flow of the sentence is a little difficult to understand without multiple readings. Can you please reword this.

Of your metabolites analyzed do they fall within traditional values for these grapes. Would you expect a satisfactory fermentation from them. 

 Line 269: I needs to be capitalized  

Author Response

Reviewer 2

Many thanks for the time you spent to correct our manuscript. Your recommendations have been followed and the corrections are highlighted in light blue marker, and they certainly improved the quality of the text. Please find all the modifications listed below.

Purpose of the paper is to use analytical and statistical approaches to evaluate Sangiovese grapes. Utilized a multidisciplinary approach looking at characteristics associated with ripening. 

Line 83 - 84: The sentence structure needs some work.

We thank the Referee. We have changed the sentence in order to improve the clarity (see lines 93-94).

Figure 1: Is it possible to provide/use a clearer picture. It's difficult to read the various cities on the map. I know you describe the differences between the different regions above and below the figure. Is it possible to give key differences about the different regions in the subtext of the figure.

We have changed Figure 1 and the caption in order to improve the clarity (lines 151-154).

What type of statistics did you run. You briefly talk about the program language. 

We thank the Referee for the comment. We have improved the information (lines 200-207).

Eq. 1: Is the equation supposed to be y = mx+b. As written I am confused by what is happening between m and t. 

Thanks for the suggestion, it was a mistake. Now it’s correct (see equation 2).

Line 248 - 251: The flow of the sentence is a little difficult to understand without multiple readings. Can you please reword this.

Thanks for the suggestion, we have changed the sentence (see lines 272-274)

Of your metabolites analyzed do they fall within traditional values for these grapes. Would you expect a satisfactory fermentation from them.

The Referee is right. The processing of the grapes was after carried out by the cellar involved int this preliminary research, showing a regular fermentation, as expected from the data obtained in this research.

Line 269: I needs to be capitalized  

We thank the Referee, it was a mistake. We have corrected the sentence (line 290).

Round 2

Reviewer 2 Report

Accept in present form